# Short Physical Performance Battery as a Measure of Physical Performance and Mortality Predictor in Older Adults: A Comprehensive Literature Review

**DOI:** 10.3390/ijerph182010612

**Published:** 2021-10-10

**Authors:** Caroline de Fátima Ribeiro Silva, Daniela Gonçalves Ohara, Areolino Pena Matos, Ana Carolina Pereira Nunes Pinto, Maycon Sousa Pegorari

**Affiliations:** Department of Biological and Health Sciences, Federal University of Amapá, Macapá 68900-000, Brazil; carolribeiro_30@hotmail.com (C.d.F.R.S.); dani_ohara@hotmail.com (D.G.O.); areolino.matos@gmail.com (A.P.M.); anacarolinapnp@hotmail.com (A.C.P.N.P.)

**Keywords:** aged, health of the elderly, longitudinal studies, mortality, survival analysis

## Abstract

The association between the Short Physical Performance Battery (SPPB) score and several adverse health outcomes, including mortality, has been reported in the scientific literature. We conducted a comprehensive literature review of studies on the relationship between SPPB and mortality. The current paper synthesizes the characteristics and main findings of longitudinal studies available in the literature that investigated the role of the SPPB in predicting mortality in older adults. The studies (*n* = 40) are from North America, South America, Europe, and Asia; the majority (*n* = 16) were conducted with community-dwelling older adults and reported an association between lower SPPB scores and a higher risk of mortality, and between higher SPPB scores and higher survival. Nevertheless, few studies have analyzed the accuracy of the instrument to predict mortality. The only study that established cut-off points was conducted with older adults discharged from an acute care hospital. Although an SPPB score lower than 10 seems to predict all-cause mortality, further studies showing cut-off points in specific settings and loco-regional specificities are still necessary.

## 1. Introduction

Over the last decades, the average life expectancy has greatly increased around the globe. In 2019, individuals aged 65 or above made up 9.1% of the worldwide population [1]. In developing countries such as Brazil, 10.8% of the population were individuals aged over 65 years in 2018, and it is expected that in 2060, older adults will comprise nearly 25.49% of the Brazilian population [2]. Nevertheless, the COVID-19 pandemic is expected to break the secular trend of increasing life expectancy [3,4,5,6]. Because of the presence of the severe acute respiratory syndrome coronavirus 2, deaths from other health conditions that were precipitated by COVID-19 and social and economic losses resulting from the pandemic are expected to be huge, meaning a rapid return to pre-COVID-19 life expectancy is unlikely [7]. Furthermore, long-term detrimental health impacts in those who recover from the virus will deserve attention, and public health policies focused on increasing the quality of life of older adults are already urgent. These policies, among several other factors, include aims to provide better understanding of the aging process and its repercussions.

At the biological level, aging results from the lifelong accumulation of cellular damage that leads to a gradual decline in physical and mental ability. Ultimately, these processes associated with aging result in increased susceptibility to conditions that lead to systemic impairments, as well as chronic diseases, frailty, sarcopenia, and a decline in physical capacity and functional ability [1,8].

Regarding physical capacity aspects, during senescence, there is a decrease in physical performance, mobility, flexibility, strength, and muscle mass [9,10,11]. Functional capacity and exercise tolerance also decrease with aging and may be associated with reduced protein synthesis and physical inactivity, or with chronic diseases, leading to functional limitations that increase the risk of disability and death [9,12].

Functional limitations and disabilities are multifactorial events, influenced by socio-demographic, clinical, and lifestyle factors [12] and therefore require early identification and preventive measures. In this regard, an important aspect in the context of geriatric clinical evaluation is the subject’s level of physical performance. Physical performance refers to the ability to integrate physiological mechanisms in coordinated movements to achieve a physical function, that is, the observable ability to perform tasks, such as getting up from or sitting on a chair [13,14].

Measures of physical performance can help to identify any risk or early stages of functional decline in older adults. Different measures are used to assess physical performance, which, in general, include mobility, balance, and muscle strength domains [15]. Using different physiological domains, these instruments usually generate a score and stratify the individual’s functional level [14]. The decline in physical performance is a dynamic and individual process. Physical performance changes according to intrapersonal alterations resulting from aging [15], and instruments for assessing physical performance have been shown to be important markers of general well-being, since they are not only parameters of mobility or strength but are also linked to the burden of chronic clinical conditions [16].

Several instruments have been cited for evaluating the physical performance of older adults. A systematic review conducted by Freiberger et al. [14] analyzed the psychometric properties of physical performance measurement tools in studies conducted with older community members, including: the Mobility-related Limitation Index (MOBLI Index), modified Timed Movement Battery (TMB), Physical Capacity Evaluation (PCE), Performance-based Physical Function test (PPF test), Physical Performance Test (PPT), and Short Physical Performance Battery (SPPB).

Two reviews [17,18] identified a number of instruments which are more frequently used to assess functional capacity and/or mobility in older adults, including the Timed Up and Go test (TUG), the 6-Minute Walk Test (6MWT), Berg Balance Scale (BBS), Shuttle Test (ST), Ergometric Test (ET), and sit to stand chair test. Of note, some instruments assess physical performance by measuring lower limb function, such as the SPPB, which has been frequently used in Brazilian studies with older adults.

The SPPB is highlighted as a diagnostic criterion for geriatric syndromes. Cesari et al. [19] in a multicenter study proposed an operational definition for physical frailty and sarcopenia using the SPPB (score ≥3 and ≤9) to detect low physical performance. The SPPB is also recommended by the European Working Group on Sarcopenia in Older People (EWGSO2) as a measure to identify declines in physical performance (SPPB score ≤ 8 points) as part of the algorithm for screening and diagnosing severe sarcopenia [20]. The Asian Working Group for Sarcopenia (AWGS) also suggests the SPPB for identifying declines in physical performance (SPPB score ≤ 9 points) as well as the 6MWT and sit to stand chair test [21].

Recent longitudinal studies have investigated multiple trajectories of physical performance measures in older adults. Hoekstra et al. [15] followed the trajectories of the physical performance of 440 subjects aged 60–70 years for 9 years, assessing balance, strength, and gait and found that there are different mechanisms involved in functional decline over time. The results of this study reinforce that, regardless of sex, physical performance incorporates individual factors (lifestyle, comorbidities, depressive symptoms, level of physical activity, among others), grouping heterogeneous aspects acquired throughout life. Mutambudzi et al. [22] followed community-dwelling older adults aged 75 years or older, also for 9 years, and classified them according to their physical performance trajectory. Participants were classified into three physical performance trajectory classes using the SPPB: low-declining, high-declining, and high-stable. The findings of the study showed a significant association between low-declining and high-declining trajectories and increased risk of mortality [22].

The burden of functional limitations and low physical performance still represents a challenging paradigm in the field of public health, and wider discussions on the health standards of the world’s populations are critical. That being said, physical performance measures are essential for not only assessing functional status, but also for monitoring the overall clinical evolution of older adults. It is worth noting that the SPPB is an easily applicable instrument, and its ability to predict adverse health events such as dependence in activities of daily living (ADLs), hospitalization, frailty syndrome, and death has been investigated in several studies conducted with community-dwelling and outpatient older populations [23,24]. However, the capacity of this tool to predict mortality and the existence of a cut-off point for discriminating older adults at risk are still little discussed. To address this gap, we conducted a comprehensive literature review of studies on the relationship between SPPB and mortality. A search using appropriate descriptors was performed in the databases MEDLINE, Embase, Lilacs, and Pedro on 22 February 2021.

## 2. Analysis of Physical Performance Using the SPPB

The SPPB assesses physical performance through balance, strength, and gait measurements and is made up of a set of three tests: standing static balance in three positions; lower limb strength and power through getting up and sitting on a chair; and walking speed at normal pace [25] (Figure 1). Balance is assessed by the ability to stand upright in three different positions for 10 seconds each: feet together; with one foot partially forward; and with one foot forward. Strength and gait are first evaluated by the ability to perform the tasks of getting up and sitting on a chair five consecutive times and performing the walking speed test (3 to 4 meters) and, second, by the time the individual takes to complete the tasks. Each test is scored from 0 (inability to perform the task) to 4 points (best test performance) [26]. The SPPB total score ranges from 0 (worst performance) to 12 points (best performance) and categorically evaluates performance in the tests using three or four classes of scores: three classes: 0–6 points (poor performance), 7–9 points (moderate performance), and 10–12 points (good performance); or four classes: 0–3 points (disability/very poor performance), 4–6 points (poor performance), 7–9 points (moderate performance), and 10–12 points (good performance) [27].

The three SPPB domains are directly related to the physical function of the older adults. The first domain is balance, which gradually decreases during senescence, mainly after the sixth decade of life [28], with a consequent decline in the ability to maintain homeostatic balance and adaptive reaction to environmental stressors. In older adults, the amplitude, frequency, and postural oscillation in the standing position is also greater than in younger subjects [28]. Declines in balance may be related to the decrease in neuromotor reactions and muscle contraction resulting from aging [29]. The SPPB assesses balance through maintenance of a static position for at least 10 seconds [25,30].

The second domain is strength. Muscle strength and power also decrease during aging and can be identified by difficulty in performing ADLs. The ability to perform ADLs is perceived in actions such as decreasing the speed at which tasks are performed or decreasing their complexity. Therefore, the functional limitation can be defined by the speed, manner, and ability to complete a task [31]. Strength is assessed in the SPPB by lower limb performance in the sit to stand chair test. Better performance in the strength test is related to less time taken to complete it, making this test essential to measure the functional capacity of older adults related to multiple daily tasks that require strength, mobility, and precision [30].

The third domain of the SPPB is gait. Walking is essential for independence in basic activities of daily living (BADL) and is an essential measure in geriatric assessments [32]. Walking speed gradually decreases with aging and at a faster pace from the age of 65, with the oldest older adults (>80 years) having a slower walking speed and shorter steps compared to younger elderly. A shorter stride length is associated with a greater decline in gait speed [33,34], and a walking speed of 0.8 m/s (meters per second) or less is a predictor of adverse clinical outcomes such as disability, cognitive decline, falls, and death [35]. As in the sit to stand chair test, better performance in the gait speed test is related to less time taken to complete the proposed task [25,30].

The SPPB was initially developed by Guralnik et al. [25] to screen older adults for the risk of disability, institutionalization, or death. The authors identified functional decline with aging and concluded that older adults with higher SPPB scores had lower functional losses compared to those with lower scores. The SPPB is a standardized and multidimensional instrument, sensitive to changes in older adult functionality [36], and that is largely associated with several health outcomes. For instance, recent longitudinal studies also found that the increase in one SPPB unit decreased the probability of falls by 15% and of recurrent falls by 17% over a two-year period. Of note, SPPB domains have not only been associated with falls [37,38,39] but also with sarcopenia [40], frailty [41], dyspnoea [42], postoperative complications [43], cardiovascular diseases [44], increased risk of mortality in chronic obstructive pulmonary disease (COPD) [45]., institutionalization, hospitalization and death [25,46,47,48]. It is also noteworthy that SPPB has been used as tool for predicting disability and physical functional impairment in discharged patients with severe COVID-19 [49].

## 3. SPPB, Mortality, and Survival in Older Adults

Studies on the association between physical performance assessed by the SPPB and mortality among older adults have been published since the 1990s. A systematic review with meta-analysis conducted by Pavasini et al. [36] analyzed the relationship between SPPB scores and all causes of mortality. The review included 17 observational studies, of which most were conducted with older people aged over 65 years. The authors analyzed all-cause mortality according to SPPB category scores. Lower SPPB scores (0–3, 4–6, and 7–9) were associated with an increased risk of death compared to higher values (scores of 10–12), and an SPPB score <10 was predictive of all-cause mortality [36].

Currently, the scientific production on longitudinal studies regarding SPPB and mortality in older adults comes from countries in North America, Europe, and Asia, with follow-ups ranging from 1 to 11 years (Figure 2). Among these studies, the majority were conducted with community-dwelling (*n* = 16) and hospitalized (*n* = 13) older adults. Only one study was conducted in South America [23], and it was conducted with older adults treated at an outpatient clinic. The characteristics of the studies are presented in Table 1.

Perera et al. [48], Rolland et al. [50], Cesari et al. [51], Legrand et al. [52], Tadjibaev et al. [53], Brown et al. [54], Fox et al. [55], Lattanzio et al. [56], Landi et al. [57], Stenholm et al. [58], Veronese et al. [59], Björkman et al. [60], and Mutambudzi et al. [22] conducted studies with community-dwelling older adults and investigated the prognostic value of the SPPB to predict mortality. In all of these studies, lower SPPB scores (range 0–6 points) significantly increased the risk of death, except for the studies by Rolland et al. [50], Verghese et al. [61], and Cesari et al. [51]. Table 2 displays the characteristics of the studies according to the SPPB classification and mortality outcome.

In the Rolland et al.’s study [50], the walking speed component was more strongly associated with mortality compared to the SPPB, with a risk ratio of 1.50 (95% CI: 0.97–2.33) versus 1.34 (95% CI: 1.04–1.73), respectively. Verghese et al. [61] reported similar results for the same variables, with a risk ratio of 1.38 (95% CI: 1.13–1.69) for walking speed versus 1.25 (95% CI: 1.06–1.47) for the SPPB score.

Cesari et al. [51] analyzed the SPPB components and found that the sit and stand test, with a risk ratio of 0.54 (95% CI: 0.38–0.76), was more strongly associated with mortality, compared to gait and balance tests, with 0.73 (95% CI: 0.54–1.01) and 0.78 (95% CI: 0.60–1.01), respectively. Verghese et al. [61] and Ensrud et al. [62] used the SPPB to assess mobility levels in community-dwelling older adults, and both studies showed an association between the lowest SPPB scores <3 and the highest risk of mortality.

In addition, some studies conducted in other settings such as with hospitalized [63,64,65,66,67,68,69,70,71,72] and institutionalized older adults [73,74]; outpatients [75]; and patients with cancer, liver injury, and those with cardiac disorders [76,77,78,79] also demonstrated an association between lower SPPB scores and an increased risk of death. In contrast, a study conducted by van Mourik et al. [80] with hospitalized older adults did not find a significant association between the SPPB and all-cause mortality (Table 2).

Regarding the survival analyses, Cesari et al. [51], Brown et al. [54], and Veronese et al. [59] conducted studies with community-dwelling older adults and found a positive relationship between SPPB scores (scores of 10-12) and survival rate, that is, older adults with better physical performance live longer when compared to those with lower performance. Similar results were found in studies with hospitalized and institutionalized older adults [63,65,73,74,81]. In the studies by Chiarantini et al. [63], Corsonello et al. [65], Charles et al. [73,74], and Arnau et al. [81], survival was significantly associated with better physical performance (SPPB scores ≥ 7), and the SPPB was found to be an independent predictor of long-term survival.

A Brazilian cohort [23] including 512 acutely ill older adults investigated the prognostic value of SPPB for dependence on basic activities of daily living—BADLs, hospitalization, and death over a one-year follow-up.

The findings were similar to international studies as they showed a higher incidence of death in patients with low (SPPB score 0–4) and intermediate (SPPB score 5–8) physical performance (risk ratio 2.70, 95% CI: 1.17–6.21, *p* = 0.042 versus 2.54; 95% CI: 1.17–5.53, *p* = 0.042) compared to patients with high performance (SPPB score ≥ 9). Figure 3 illustrates the association of SPPB scores with mortality and survival.

## 4. SPPB Accuracy for Predicting Mortality

Some studies analyzed the area under the ROC curve (Receiver Operating Characteristic Curve—AUC), and cut-off points were established to verify the accuracy of the SPPB to predict mortality in older adults from different settings. Three studies investigated the accuracy of the SPPB to predict mortality in community-dwelling older adults. Minneci et al. [82] compared the capacity of physical performance tests, including the SPPB, to predict mortality and other clinical outcomes among 561 older adults over a 7-year period. The SPPB was shown to be a better predictor of mortality compared to other measures of performance, with an area under the ROC curve of 0.63.

Landi et al. [57] verified the impact of sarcopenia and its relationship with functional decline on the risk of all-cause mortality in 354 community-dwelling older adults during a 10-year follow-up. Impairment in physical function in sarcopenic older adults as assessed by the SPPB was found to be a better predictor of mortality (AUC: 0.697; 95% CI: 0.639–0.755) than multimorbidity (AUC: 0.633; 95% CI: 0.572–0.695).

Cesari et al. [51] analyzed the predictive ability of the SPPB combined with self-rated health status during a 24-month follow-up and did not find significant differences between ability in the sit to stand chair test (AUC: 0.725; 95% CI: 0.661–0.789), self-rated health (AUC: 0.656; 95% CI: 0.582–0.730), and their combination (AUC: 0.751; 95% CI: 0.686–0.816) to predict mortality (AUC: 0.749; 95% CI: 0.683–0.814), and they reported similar results for the isolated analysis of SPPB scores (AUC: 0.743; 95% CI: 0.679–0.806). On the other hand, in 2013, the same authors measured the prognostic value of multiple screening tools for the assessment of 1-year mortality risk in 200 older women with gynecological cancer and found only borderline significance for the SPPB (AUC: 0.638; 95% CI: 0.483–0.792) in predicting mortality [83].

Two studies analyzed the association of the SPPB with all-cause mortality in older adults with cardiac disorders. Afilalo et al. [84] investigated the value of frailty scales (including SPPB) to predict one-year mortality in older adults undergoing surgical or transcatheter aortic valve replacement. The findings showed that SPPB is not the best scale to predict mortality (AUC: 0.734; 95% CI: 0.694−0.775) compared to other scales used to identify frailty. Campo et al. [85] described that the SPPB combined with the GRACE (Global Registry of Acute Coronary Events) (AUC:0.816, 95% CI: 0.777–859) and TIMI (Thrombolysis in Myocardial Infarction) (AUC: 0.879, 95% CI: 0.814–0.884) risk scores provided incremental improvements in risk stratification for death in 1 year of older adults after acute coronary syndrome.

Of note, only one study established cut-off points for the SPPB to predict mortality. Corsonello et al. [65] investigated the prognostic role of the SPPB to predict survival and mortality during a 1-year follow-up in 506 older adults discharged from an acute care hospital. The results showed that a score <5 in the SPPB was capable of predicting mortality (AUC: 0.66), with sensitivity and specificity values of 0.66 and 0.62, respectively.

## 5. Conclusions

The SPPB is an easily applicable and low-cost instrument that may be implemented in the routine health assessment of older adults for screening geriatric clinical conditions. It is associated with falls, sarcopenia, frailty, dyspnoea, postoperative complications, cardiovascular diseases, institutionalization, and ultimately, death. This research provides important information upon which to base future primary health care policies for older people aiming at preventing adverse health outcomes, especially death. Although an SPPB score lower than 10 seems to predict all-cause mortality, different configurations of SPPB scoring categories in diverse services and health settings could also provide predictive power for this outcome. Thus, further studies demonstrating cut-off points in specific settings and loco-regional specificities are still necessary.

## Figures and Tables

**Figure 1 ijerph-18-10612-f001:**
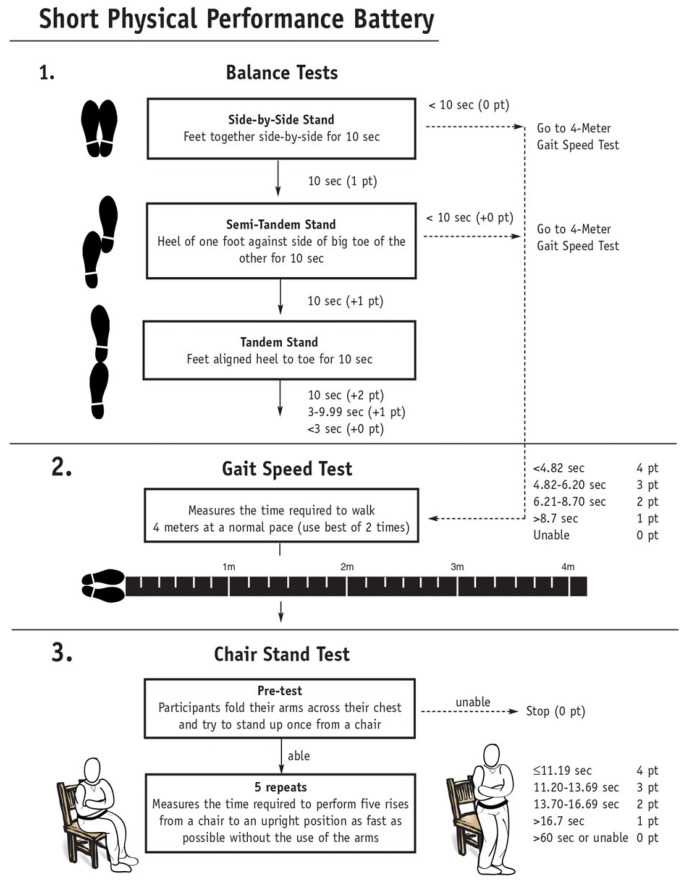
Short physical performance battery—SPPPB. Source: Wall chart courtesy of Dr. Jack Guralnik.

**Figure 2 ijerph-18-10612-f002:**
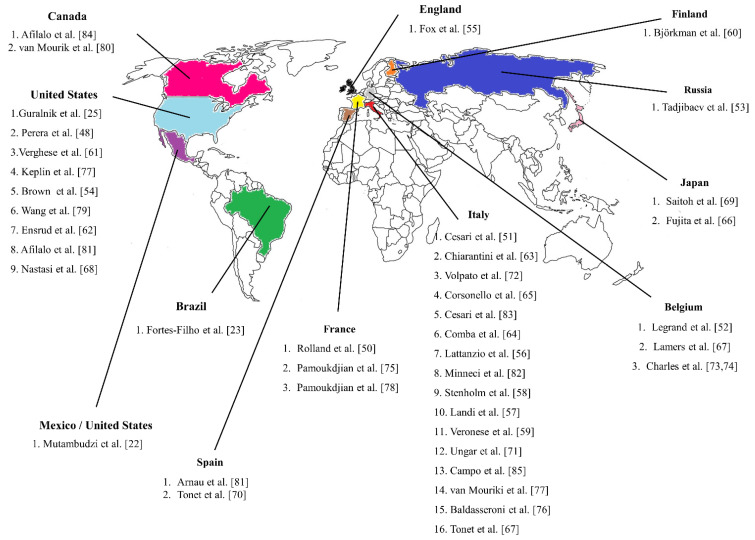
Distribution of longitudinal studies conducted with older adults on the SPPB and mortality according to research locations.

**Figure 3 ijerph-18-10612-f003:**
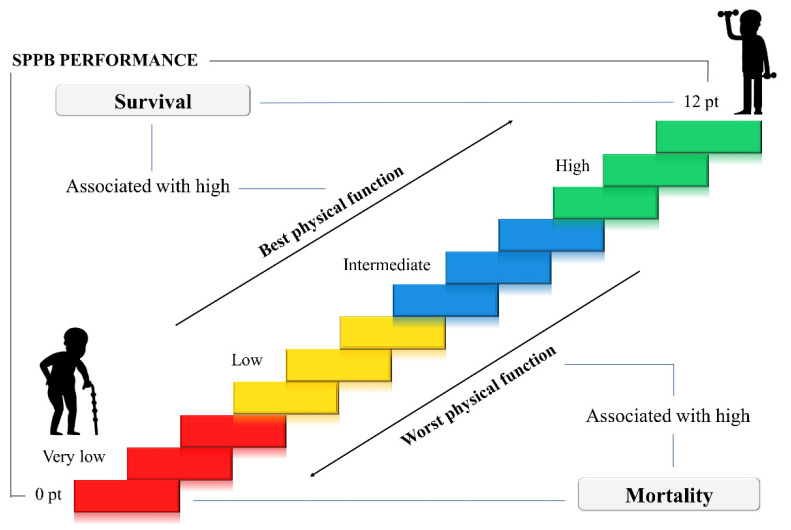
Association of mortality and survival according to SPPB categories (very low 0–3, low 4–6, intermediate 7–9, high 10–12).

**Table 1 ijerph-18-10612-t001:** Characteristics of longitudinal studies conducted with older adults on the relationship between the SPPB and mortality (*n* = 40).

Authors	Country	Sample/Setting	Follow-Up
Mutambudzi et al. [22]	Mexico/United States	1411 community-dwelling older adults	9.5 years
Fortes-Filho et al. [23]	Brazil	512 acutely ill older outpatients	1 year
Guralnik et al. [25]	United States	5174 community-dwelling and institutionalized older adults	6 years
Perera et al. [48]	United States	439 community-dwelling older adults	5 years
Rolland et al. [50]	France	7250 community-dwelling older women	3.8 years
Cesari et al. [51]	Italy	335 community-dwelling older adults	24 months
Legrand et al. [52]	Belgium	560 community-dwelling older adults	33.5 months
Tadjibaev et al. [53]	Russia	284 community-dwelling older adults	2.6 years
Brown et al. [54]	United States	413 older adult cancer survivors	11 years
Fox et al. [55]	England	213 older adults living in suburban and urban sectors	4 years
Lattanzio et al. [56]	Italy	487 community-dwelling older patients discharged from acute care hospitals	1 year
Landi et al. [57]	Italy	364 community-dwelling older adults	10 years
Stenholm et al. [58]	Italy	996 community-dwelling older adults	10 years
Veronese et al. [59]	Italy	2096 community-dwelling older adults	4.4 years
Björkman et al. [60]	Finland	428 community-dwelling older adults	4 years
Verghese et al. [61]	United States	631 community-dwelling older adults	32 months
Ensrud et al. [62]	United States	1495 community-dwelling older women	4.9 years
Chiarantini et al. [63]	Italy	157 older subjects hospitalized for decompensated heart failure	30 months
Comba et al. [64]	Italy	1621 hospitalized older adults	7 months
Corsonello et al. [65]	Italy	506 older adults discharged from an acute care hospital	1 year
Fujita et al. [66]	Japan	147 hospitalized older adults	1 year
Lamers et al. [67]	Belgium	302 hospitalized older adults	4 years
Nastasi et al. [68]	United States	142 hospitalized older adults	5 years
Saitoh et al. [69]	Japan	463 hospitalized older adults	3 years
Tonet et al. [70]	Italy/Spain	908 hospitalized older adults	288 days
Ungar et al. [71]	Italy	71 hospitalized older adults	3 months
Volpato et al. [72]	Italy	87 hospitalized older adults	3 months
Charles et al. [73,74]	Belgium	604 institutionalized older adults	3 years
Pamoukdjian et al. [75]	France	603 older adults with cancer	6 months
Baldasseroni et al. [76]	Italy	235 hospitalized older adults	5 years
Klepin et al. [77]	United States	74 older adults with acute myelogenous leukaemia	30 days
Pamoukdjian et al. [78]	France	190 older adults with cancer	6 months
Wang et al. [79]	United States	95 older adults on the liver transplant waiting list	14 months
van Mourik et al. [80]	Italy/Netherlands/Canada	71 hospitalized older adults	1 year
Arnau et al. [81]	Spain	315 of the oldest old population attending primary care	10 years
Minneci et al. [82]	Italy	561 community-dwelling older adults	7 years
Cesari et al. [83]	Italy	200 older women with gynaecological cancer	1 year
Afilalo et al. [84]	Canada/United States/France	1020 older adult patients undergoing surgical or transcatheter aortic valve replacement	1 year
Campo et al. [85]	Italy	402 hospitalized older adults	1 year

Source: Authors.

**Table 2 ijerph-18-10612-t002:** Characteristics of longitudinal studies according to the SPPB classification and mortality outcome (*n* = 40).

Authors	Age Range (years)	SPPB Classification	Mortality Results
Mutambudzi et al. [22]	81.1 ± 4.5	Three trajectory classes of SPPB scores (low declining, high declining, and high stable)	High-declining physical performance—HR: 1.64 (1.32–2.03)
Fortes-Filho et al. [23]	79.4 ± 8.3	Low (0–4), intermediate (5–8), and high (9–12) performance	Low (0–4)—HR: 2.70 (1.17–6.21), intermediate (5–8)—HR: 2.54 (1.17–5.53)
Guralnik et al. [25]	>71	SPPB scores 0–12, low (≤5) and high performance (8–12)	Low ≤ 5, men—HR: 2.3 (1.8–2.9), women—HR: 2.6 (2.0–3.5)
Perera et al. [48]	73.9 ± 5.6	SPPB score—continuous variable	SPPB score persistently declined in 5 years (1 point change)—HR: 2.48 (1.36–4.50)
Rolland et al. [50]	80.5 ± 3.76	Low (0–6), intermediate (7–9), and high performance (10–12)	Low (0–6)—HR: 1.50 (0.97–2.33)
Cesari et al. [51]	85.6 ± 4.8	SPPB score—continuous variable	SPPB score—HR 0.64 (0.48–0.86)
Legrand et al. [52]	84.7 ± 3.7	Women—low (0–5), intermediate (6–8), and high performance (9–12)/men—low (0–7), intermediate (8–10), and high performance (11–12)	SPPB highest tertiles were associated with less risk of death than the lowest tertiles—HR: 0.68 (0.48–0.98)
Tadjibaev et al. [53]	70.7 ± 2.3 (65–74) 79.8 ± 3.4 (<75)	SPPB score—continuous variable	Poor physical performance (SPPB score) aged 65–74—HR: 2.1 (0.59–7.7) and aged >75 HR: 4.2 (1.5–11.5)
Brown et al. [54]	72.2 ± 0.47	Low (0–6), intermediate (7–9), and high performance (10–12), SPPB score—continuous variable	Intermediate (7-9) predicted reduction in mortality—HR: 0.57 (0.37–0.89) and high performance (10-12)—HR: 0.50 (0.32–0.77) SPPB score (1-unit increase) predicted 12% reduction in mortality—HR: 0.88 (0.82–0.94)
Fox et al. [55]	>70	Low (≤6), intermediate (7–9), and high performance (10–12)	Low (≤6)—HR: 5.30 (1.91–14.72) and intermediate (7-9)—HR: 2.58 (0.89–7.52)
Lattanzio et al. [56]	80.1 ± 6.0	Low (0–4), intermediate (5–8), and high performance (9–12)	Low (0-4)—HR: 2.93 (1.07–8.63)
Landi et al. [57]	84.2 (range 80–102)	Very low (0–2), low (3–5), moderate (6–8), and high performance (≥9), SPPB score to analyze physical function in sarcopenic older adults	Higher levels of physical function (SPPB score ≥ 9) were associated with longer survival in sarcopenic older adults
Stenholm et al. [58]	Men 74.0 ± 7.0 Women 75.4 ± 7.5	SPPB score classified (inactive, Moderate, and active)	Inactive—HR: 1.73 (0.78–3.82) and moderate—HR: 1.26 (0.57–2.79)
Veronese et al. [59]	75.2 ± 6.1	SPPB score—continuous variable	Two lowest quartiles of SPPB tests—HR: 2.06 (1.27–3.34) and HR: 1.84 (1.10–3.05)
Björkman et al. [60]	83.4 ± 4.6	SPPB score—continuous variable	SPPB score—HR: 0.85 (0.79–0.72)
Verghese et al. [61]	79.9 ± 5.3	SPPB score—continuous variable	SPPB score (1 point change)—HR: 1.25 (1.06–1.47)
Ensrud et al. [62]	87.6 ± 3.3	Low (0–3), intermediate (4–9), and high performance (10–12)	Low (0–3)—HR: 1.64 (1.24–2.16), intermediate (4–9)—HR: 1.26 (1.02–1.57)
Chiarantini et al. [63]	80 ± 0.5	Incapacity (0), low (1–4), intermediate (5–8), and high performance (9–12)	Incapacity (0)—HR: 6.06 (2.19–16.76), low (1–4) —HR: 4.78 (1.63–14.02), and intermediate (5–8)—HR: 1.95 (0.67-5.70)
Comba et al. [64]	82.0 ± 7.7	Low (0–6), intermediate (7–10), high (11–12)	Low (0-6)—OR: 0.43 (*p* = 0.050)
Corsonello et al. [65]	80.1 ± 5.9	Low (0–4), intermediate (5–8), and high performance (9–12)	Intermediate (5-8)—HR: 0.76 (0.40–1.68) and high (9–12)—HR: 0.51 (0.30-1.05)
Fujita et al. [66]	86.5 ± 4.7	Incapacity (0), low (1–6), and high performance (7–12) SPPB score—continuous variable	SPPB score, low—HR: 0.41 (0.22–0.79) and high—HR: 0.26 (0.12–0.58)
Lamers et al. [67]	85.9 ± 6.3	Low (0–4), intermediate (5–7), and high performance (8–12)	Mortality risk higher 59.3% in low score (0–4) compared to high score (8–12)—HR: 0.40 (0.23–0.70) and intermediate—HR: 0.44 (0.29–0.67)
Nastasi et al. [68]	Group ≥ 65	SPPB score—impairment (<10)	SPPB impairment group—HR: 2.60 (1.00–6.80)
Saitoh et al. [69]	85 (range 82–88)	SPPB score—continuous variable	SPPB score (1-unit decrease)—OR 2.10 (1.11–3.96)
Tonet et al. [70]	82 ± 6	SPPB score—continuous variable	Lower SPPB scores—HR: 0.88 (0.82–0.95)
Ungar et al. [71]	85.4 ± 2.9	SPPB score—continuous variable	Mortality or hospitalization risk in participants with low SPPB scores: OR: 1.15 (1.01–1.54); mortality or non-fatal stroke risk in participants with low SPPB scores—OR: 1.62 (1.08–2.43)
Volpato et al. [72]	77.4 (range 65–93)	Low (0–4), intermediate (5–7), and high performance (8–12)	Low (0-4)—OR: 5.38 (1.82-15.9)
Charles et al. [73,74]	82.9 ± 9.1	SPPB score tests (balance, gait speed, and sit to stand chair)—continuous variable SPPB score (fast decline and moderate decline)—continuous variable	Balance—HR: 0.88 (0.78–0.99), gait speed—HR: 0.89 (0.76–1.03), and sit to stand chair—HR: 0.97 (0.82–1.15) Fast decline—HR: 1.78 (1.34–2.26) and moderate decline—HR: 1.37 (1.10–1.66)
Pamoukdjian et al. [75]	81.2 ± 6.1	Impaired mobility (<9), normal mobility (≥9)	SPPB score (<9)—HR: 3.03 (1.93–4.76)
Baldasseroni et al. [76]	79.6 ± 0.2	SPPB score—impairment (<7)	Mortality risk higher postoperative—OR 0.77 (0.66–0.89)
Klepin et al. [77]	70 ± 6.2	SPPB score—continuous variable, low performance (<9), high performance (≥9)	SPPB score (2 point increase reduced the risk of death by 15%)—HR: 0.85 (0.72–1.01), low (<9)—HR: 2.2 (1.1–4.6)
Pamoukdjian et al. [78]	80.6 ± 5.6	SPPB score—impairment (<9)	SPPB score—HR: 5.8 (1.6–20.9)
Wang et al. [79]	67 (range 66–69)	SPPB score—continuous variable	SPPB score (1-unit) ≥ 9—HR = 1.57 (0.81–3.05) and <9—HR= 2.36 (1.19–4.66)
van Mourik et al. [80]	85.4 ± 2.9	High risk (0–6), low risk (7–12)	High risk (0–6)—OR: 7.09 (0.70–71.89)
Arnau et al. [81]	81.9 ± 4.7	SPPB score low (<7) and high performance (≥7)	Mortality risk 10-years score (<7)—0.23 and (≥7) —0.37; survival 10-years, SPPB score <7—HR: 1.37 (1.01–1.86)
Minneci et al. [82]	72.9 ± 0.3	SPPB score—continuous variable	SPPB score (1-unit)—HR: 0.92 (0.85– 0.99)
Cesari et al. [83]	73.5 ± 6.2	SPPB score—continuous variable	SPPB score—HR: 0.54 (0.29–0.98)
Afilalo et al. [84]	82 (77–86)	SPPB score—continuous variable	Mortality 30 days after cardiac procedure—OR: 4.07 (1.43–11.60) and 1 year—OR: 2.96 (1.75–5.00)
Campo et al. [85]	78 ± 6	SPPB score—continuous variable	SPPB score—OR: 0.74 (0.63–0.85)

Mean ± standard deviation; median (interquartile interval); HR: hazard ratio (95% CI: confidence interval); OR: odds ratio. Source: Authors.

## Data Availability

Not applicable.

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
