# Peer review of "Short Physical Performance Battery as a Measure of Physical Performance and Mortality Predictor in Older Adults: A Comprehensive Literature Review"

_ijerph, 2021, doi:10.3390/ijerph182010612_

Round 1

Reviewer 1 Report

Manuscript IJERPH-1388257; Short physical performance battery as a measure of physical performance and mortality predictor in older adults: a comprehensive literature review.

The manuscript presents an original and interesting issue. In my opinion, it has an important clinical relevance and has valuable implications in the management of older adults. Moreover, the manuscript properly synthesizes the literature regarding SPPB.

The writing and sequencing of the information is adequate in this manuscript, and the content is clear and properly presented. Reading it is easy and fluent, which makes this paper highly recommended for researchers interested in this topic.

Nevertheless, there is one important issue that need to be pointed.

Although the references list includes very recent literature, which increases the value of this paper, in my opinion authors should consider including a new section in the manuscript regarding the relationship between SPPB and some aging-related diseases (and not only the link with mortality), such as cardiovascular disease and pulmonary disease, which could lead to a higher mortality in older adults.

In addition, since authors highlight the impact of the COVID-19 in the mortality rate in this population, it could be interesting to explain the role of the SPPB as a mortality predictor in people affected by the COVID-19 and other important diseases. In fact, recent literature regarding this topic has been published in the last years. 

Author Response

Point 1: The manuscript presents an original and interesting issue. In my opinion, it has an important clinical relevance and has valuable implications in the management of older adults. Moreover, the manuscript properly synthesizes the literature regarding SPPB.

The writing and sequencing of the information is adequate in this manuscript, and the content is clear and properly presented. Reading it is easy and fluent, which makes this paper highly recommended for researchers interested in this topic.

Nevertheless, there is one important issue that need to be pointed.

Although the references list includes very recent literature, which increases the value of this paper, in my opinion authors should consider including a new section in the manuscript regarding the relationship between SPPB and some aging-related diseases (and not only the link with mortality), such as cardiovascular disease and pulmonary disease, which could lead to a higher mortality in older adults.

Response 1: We are thankful for the observations of the reviewer. Although the relevance of the comment on review, the central idea of ​​this review is about SPPB and its predictive capacity for mortality in older adults, so we chose for the inclusion of such information succinctly in the topic 2 (lines 170 and 171). In this topic, the authors mentioned SPPB and its association with adverse outcomes in older adults, as well as frailty, sarcopenia and falls.

Point 2: In addition, since authors highlight the impact of the COVID-19 in the mortality rate in this population, it could be interesting to explain the role of the SPPB as a mortality predictor in people affected by the COVID-19 and other important diseases. In fact, recent literature regarding this topic has been published in the last years.

English language and style are fine/minor spell check required.

Response 2: The authors are grateful for the relevant comments. As previously mentioned, we cited in the text the relationship of SPPB and mortality, such as SPPB and geriatric syndromes and diseases. Specifically, about Covid-19, we included information succinctly in topic 2 (lines 172 and 173), emphasizing that SPPB has been used as tool for predicting disability and physical functional impairment   in discharged patients with severe Covid-19. We also inform the reviewer that English language was properly revised in the manuscript, and the certificate was attached on submission process.

Reviewer 2 Report

This is a well-prepared manuscript looking at the literature on SPPB in older adults. It is well-prepared and almost ready for publication.

Introduction

First sentence: yes, everyone has been aging since we became humans thousands of years ago. Do you mean people are living longer? Please updating the first sentence to reflect the meaning intended

Conclusions

Need to strengthen the conclusions.

I would have liked to have seen more detail and more conclusion material.

Author Response

Point 1: Introduction

First sentence: yes, everyone has been aging since we became humans thousands of years ago. Do you mean people are living longer? Please updating the first sentence to reflect the meaning intended

Response 1: We are grateful for the comments of the reviewer. We proceeded with the appropriate corrections in the manuscript, topic introduction (lines 27 and 28).

Point 2: Conclusions

Need to strengthen the conclusions.

I would have liked to have seen more detail and more conclusion material.

English language and style are fine/minor spell check required.

Response 2: We thank again by the important observation on our manuscript, and we included the necessary information, topic conclusions (lines 284 – 293). We also inform the reviewer that English language was properly revised in the manuscript, and the certificate was attached on submission process.